# Occurrence and Risk Assessment of Personal PM_2.5_-Bound Phthalates Exposure for Adults in Hong Kong

**DOI:** 10.3390/ijerph192013425

**Published:** 2022-10-18

**Authors:** Jiayao Chen, Tony J. Ward, Steven Sai Hang Ho, Kin Fai Ho

**Affiliations:** 1Department of Real Estate and Construction, The University of Hong Kong, Hong Kong SAR, China; 2Shenzhen Institute of Research and Innovation, The University of Hong Kong, Shenzhen 518057, China; 3School of Public and Community Health Sciences, University of Montana, Missoula, MT 59801, USA; 4Division of Atmospheric Sciences, Desert Research Institute, Reno, NV 89512, USA; 5The Jockey Club School of Public Health and Primary Care, The Chinese University of Hong Kong, Hong Kong SAR, China

**Keywords:** personal exposure, residential indoor, inhalation exposure, composition, di(2-ethylhexyl) phthalate (DEHP), risk assessment

## Abstract

We performed personal PM_2.5_ monitoring involving 56 adult residents in Hong Kong. Additionally, paired personal and residential indoor fine particle (PM_2.5_) samples were collected from 26 homes and from 3 fixed monitoring locations (i.e., outdoor samples). Six PM_2.5_-bound phthalate esters (PAEs)—including dimethyl phthalate (DMP), diethyl phthalate (DEP), di-n-butyl phthalate (DnBP), butyl benzyl phthalate (BBP), di(2-ethylhexyl) phthalate (DEHP), and di-n-octyl phthalate (DnOP)—were measured using a thermal desorption–gas chromatography/mass spectrometer method. Average ∑_6_PAEs (i.e., summation of six PAE congeners) concentrations in personal PM_2.5_ exposure (699.4 ng/m^3^) were comparable with those in residential indoors (646.9 ng/m^3^), and both were slightly lower than the outdoor levels. DEHP was the most abundant PAE congener (80.3%–85.0%) and found at the highest levels in different exposure categories, followed by BBP, DnBP, and DnOP. Strong correlations were observed between DEHP with DnBP (*r*_s_: 0.81–0.90; *p* < 0.01), BBP (*r*_s_: 0.81–0.90; *p* < 0.01), and DnOP (*r*_s_: 0.87–0.93; *p* < 0.01) in each exposure category. However, no apparent intercorrelations were shown for PAE congeners. Higher indoor concentrations and a stronger correlation between DMP and DEP were found compared with outdoor concentrations. Principal component analysis affirmed heterogeneous distribution and notable variations in PAE sources across different exposure categories. The average daily intakes of ∑_6_PAEs and DEHP via inhalation were 0.14–0.17 and 0.12–0.16 μg/kg-day for adults in Hong Kong. A time-weighted model was used to estimate PAE exposures incorporating residential indoor and outdoor exposure and time activities. The inhalation cancer risks attributable to measured and estimated personal exposure to DEHP exceeded the U.S. EPA’s benchmark (1 × 10^−6^). The results provide critical information for mitigation strategies, suggesting that PAEs from both ambient and indoor sources should be considered when exploring the inhalation health risks of PAEs exposure.

## 1. Introduction

“Walking into a modern building can sometimes be compared to placing your head inside a plastic bag that is filled with toxic fumes” (John Bower, the founder of the Healthy House Institute). Phthalate esters (PAEs) are a group of synthetic chemicals widely used in polyvinyl chloride (PVC) products and consumer products (such as commodities, medical products, cosmetics, and personal care products) and in households (building materials, furnishing, household goods) [1,2]. Due to their widespread use, the negative impacts of PAEs on human health have raised global concerns [3,4]. PAEs are ubiquitous chemical contaminants detected in various environmental matrices, including air, water, soil, sediments, and sludge [5,6]. Several PAE congeners—including dimethyl phthalate (DMP), diethyl phthalate (DEP), di-n-butyl phthalate (DnBP), di-iso-butyl phthalate (DiBP), butyl benzyl phthalate (BBP), di(2-ethylhexyl) phthalate (DEHP), and di-n-octyl phthalate (DnOP)—have been investigated due to their potential to cause adverse health effects following exposure [7]. The International Agency for Research on Cancer and the United States Environmental Protection Agency (U.S. EPA) have classified DEHP and BBP as possible human carcinogens (Group B2 and Group C) based on experimental animal studies [8,9]. Epidemiological studies have demonstrated that exposure to PAEs has been associated with childhood obesity [10,11], male developmental issues and reproductive outcomes [12,13], and female breast cancer incidence [14,15]. Moreover, studies have found associations between inhalation exposure to PAEs and the development of asthma and allergic symptoms in children [16,17]. In addition, exposure to PAEs has been linked to potential effects on human respiratory problems and blood pressure in adults. 

Characteristics of common air pollutants and air toxics (e.g., benzene, formaldehyde, volatile organic compounds, vinyl chloride, metals, and persistent organics) have been investigated in previous studies [18,19]. Studies have indicated that PAEs are the most abundant endocrine-disrupting chemicals in ambient air in different countries [20,21,22]. For example, an investigation conducted in urban sites across 16 Chinese cities found that PM_2.5_-bound PAE concentrations in ambient air ranged from 63 to 1162 ng/m^3^ [20]. Given that a large proportion (>85%) of daily time is spent indoors for the general population, research efforts have been made to address PAEs in different indoor microenvironments, including schools, offices, and residential homes [23] and from indoor dust [24]. Buildings offer partial protection again ambient origin particulate pollutants, but indoor sources of PAEs enhance the potential for overall personal exposures. PAEs are physically bound to the plastic polymer and can be easily released into the ambient atmosphere and adhere to indoor particles and settled house dust [16,25]. 

In addition to ingestion and dermal contact, a growing body of evidence shows that air inhalation is an important route of nonoccupational exposure to PAEs for adults and children [26]. Characteristics of exposure to metabolites of phthalates (e.g., DEP, DnBP, and DEHP were the most abundant compounds) in human biological samples (e.g., urine, blood serum) have also been investigated [27]. However, few studies have investigated personal exposure to particle-bound PAEs at the individual level. How well these measurements may represent personal exposures in the long term remains to be answered. 

The research objectives of this study are to (1) examine the occurrence, concentrations, and variations of PM_2.5_-bound PAE congeners (i.e., DMP, DEP, DnBP, BBP, DEHP, and DOP) in personal exposure and residential indoors; (2) characterize the within- and between-individual (or home) variability of PAE congeners in personal exposure and residential indoor; (3) investigate the potential sources of PAEs in different exposure metrics; and (4) assess health risks caused by inhalation exposure to DEHP for Hong Kong adults based on directly measured exposure and estimated personal exposure. This investigation provides new information regarding the inhalation risks of personal exposure to particle-phase PAE indoors and outdoors in the general population, leading to targeted and effective mitigation strategies in populated urban areas. 

## 2. Materials and Methods

### 2.1. Participants and Study Design 

We used a random sampling strategy to recruit potential subjects without restricting gender or occupation. Figure 1a illustrates the study design using a flowchart. Characteristics of the study subjects have been described in detail in another publication [28]. In brief, 79 adults were recruited, and 71% (56) agreed to participate in the personal PM_2.5_ measurement in the summer and winter seasons between June 2014 and March 2016 [29].

### 2.2. Sample Collection 

Personal exposure, residential indoor, and outdoor PM_2.5_ samples were collected as described in a previous publication [28]. Individual PM_2.5_ samples were collected using a personal environmental monitor (PEM) connected to a Leland Legacy pump. The samplers were operated at a flow rate of 10 (±0.5) L/min for 24 hours (24 h). Personal samplers were attached to the breathing zone of study participants. Each subject carried two samplers, one PEM loaded with quartz and the other with a Teflon filter. Two distinct seasons were investigated: June to September for summer and November to March for winter. Personal samples were collected repeatedly for two days during summer and winter. One hundred eighty-four sets of personal PM_2.5_ samples were obtained throughout the study period.

Of the 56 study participants, 26 participated in contemporary personal and residential indoor PM_2.5_ sample collection (Figure 1a). Two parallel Mini-Volume air samplers were placed at ~1.5 m above the ground in the participants’ living room and operated at a flow rate of 5 (±0.25) L/min for 24 h. Residential indoor measurements were performed during the same sampling period as personal exposure. As a result, two to four samples were obtained from each household, leading to 63 sets of indoor PM_2.5_ samples throughout the study period. In addition, outdoor PM_2.5_ samples were collected concurrently at fixed sites in the same district as residential indoors, with 63 sets of outdoor PM_2.5_ samples collected (Figure 1b). More details and a discussion concerning the three outdoor locations are described in [28] and the Appendix A.

### 2.3. Phthalates Analysis 

The analytical methods utilized in this study are described in a previous publication [29]. In brief, a punch (0.526–2.630 cm^2^) of the exposed quartz filter was subjected to particle-phase organic compound determination (including phthalates and polycyclic aromatic hydrocarbons (PAHs)), employing the thermal desorption–gas chromatography/mass spectrometer (TD–GC/MS) method [30]. Our findings for health risks of PAHs were reported separately [28]. 

In this investigation, the targeted chemical species consist of six particle-phase PAE congeners, including dimethyl phthalate (DMP), diethyl phthalate (DEP), di-n-butyl phthalate (DnBP), butyl benzyl phthalate (BBP), di(2-ethylhexyl) phthalate (DEHP), and di-n-octyl phthalate (DnOP). PAE congeners returned concentrations exceeding method detection limits (MDLs) for >93.8% of the personal PM_2.5_ samples, and indoor PAEs were detected in 91.5% of residential locations (Appendix A). DEHP and BBP were detected >MDLs in 86.9%–94.6% of PM_2.5_ samples. PAEs on filter samples were detected along with blanks, and the reported PAE congener concentrations were corrected by subtracting average blank values. Sample preparation (e.g., quartz filters preheated at 900 °C for 3 h), after sampling storage (e.g., at –4 °C), chemical analyses, and the corresponding quality assurance/quality control protocols have been reported previously [28]. 

### 2.4. Risk Assessment of Inhalation Exposure to PAEs

Exposure concentrations (EC, ng/m^3^) of PAE congeners attributable to inhalation were estimated using the following equation [31]:(1)$${\mathrm{EC}}_{i}=\frac{C_{i}\times \mathrm{ET}\times \mathrm{EF}\times \mathrm{ED}}{\mathrm{AT}}$$
where C*_i_* represents the concentration of the studied PAE congener (*i*) in PM_2.5_ samples (ng/m^3^), and ET is the exposure time (24 h/day). EF is exposure frequency (e.g., calculated based on the average frequency of detection for PAEs in personal, residential indoor, and outdoor samples, with results showing 342 year, 334, and 341 days/year, respectively (Appendix A). ED is the exposure duration in years (e.g., 70 years for adults). For carcinogenic pollutants, AT is the average time in hours (i.e., ED × 365 days/year × 24 h/day) [32,33]. 

In addition, we calculated the daily intake (DI*_inh_*, μg/kg-day) of PAEs in air samples as follows:(2)$${\mathrm{DI}}_{i\_inh}=\frac{{\mathrm{EC}}_{i}\times {\mathrm{IR}}_{inh}\times \mathrm{EF}\times \mathrm{ED}}{\mathrm{BW}\times T\times \mathrm{CF}}$$
where T is the averaging time in days (i.e., ED × 365 days/year), CF is the conversion factor (1000 ng/μg), BW is the body weight (kg) of individual participants, and *IR_inh_* is the participants’ inhalation rate (m^3^/day) that was estimated, incorporating the *Exposure Factors Handbook* for specific personal activities from adults [33]. In addition, general information of study participants (e.g., gender, body weight) was extracted from the questionnaires and activity diaries. 

In this research, potential cancer risk attributable to inhalation exposure to DEHP was calculated using the following equation: (3)$${\mathrm{CR}}_{inh}={\mathrm{DI}}_{inh}\times {\mathrm{CSF}}_{inh}$$
where CSF*_inh_* refers to the cancer slope factor for DEHP. The cancer potency value for DEHP was adapted from the new risk assessment algorithms defined in the OEHHA (California Office of Environmental Health Hazard Assessment) Air Toxic Hot Sports Program [34], with an inhalation cancer potency factor of 0.0084 (mg/kg-day)^−1^ used for risk assessment in the current study. Other PAE congeners are not further discussed because their CSF*_inh_* values are not available. 

### 2.5. Literature Review and Studies Comparison 

We performed a literature review to identify studies that investigated the concentrations of PM_2.5_-bound PAEs across different exposure metrics, including personal exposures, indoor microenvironments, and ambient (outdoor) environments. We searched Web of Science, PubMed, and Scopus for English-language peer-reviewed journal articles from database inception until December 2021. We identified the targeted studies employing the search strategy with keywords: (“phthalate*” OR “phthalate ester*” OR “phthalic acid ester” OR “endocrine disruptor*” OR “endocrine disrupting chemical*” OR “phthalate congener*”) AND (“fine particle” OR “fine particulate matter” OR PM2.5 OR PM 2.5 OR PM25 OR PM 25) AND (“personal exposure” OR “individual exposure” OR “indoor*” OR “residential indoor*” OR “outdoor” OR “ambient”). Searches of all three databases and retrieval of results from the full texts were completed on 15 January 2022. The detailed search strategy is shown in Appendix A. Eligible studies were included during the screening if: (1) the observational studies were conducted in individual participants, indoors or ambient (outdoor); (2) studies focused on the analyzed six PAE congeners (DMP, DEP, DnBP, BBP, DEHP, DnOP) in PM_2.5_; (3) concentrations of PAE congeners (ng/m^3^) were reported; and (4) studies were published in peer-reviewed journals in English. Appendix A shows that 132 records were obtained from the searched databases. After removing duplicates (n = 28) and reviewing articles (n = 8), 96 papers were assessed at the titles/abstracts level, and 55 full texts were reviewed. Finally, 35 studies concerning PM_2.5_-bound PAEs were included for comparison purposes, including 20 in ambient (outdoor), 15 indoors/personal, and 5 on personal exposures. This investigation compared the personal exposure and residential indoor PAE concentrations with the extracted results. Outdoor measures and literature review concerning these priority PAEs in ambient (outdoor) PM_2.5_ in this publication are listed in the Appendix A. 

### 2.6. Source Identification 

Average indoor-to-outdoor (I/O) ratios for PAEs were calculated. We used Spearman’s correlation coefficients (*r*_s_) to characterize the associations of PAE congeners in and between exposure categories. In addition, we applied principal component analysis (PCA) to identify the potential sources of PAEs in personal exposure, residential indoor, and outdoor PM_2.5_. This method has been employed for indoor and outdoor PAEs source identification [35] and personal exposure source apportionment in a previous investigation [36]. In this study, we used varimax normalized rotation to minimize or maximize loading factors of included species for each rotated principal component. The principal components with eigenvalues greater than 1 were retained, and a factor loading of less than 0.4 was omitted to facilitate source identification. PCA was performed by using IBM SPSS Statistics (Version 26.0, Armonk, NY, USA: IBM Corp). 

### 2.7. Statistical Analysis 

PAE congener concentrations are reported in ng/m^3^. The Shapiro–Wilk test is used to check the normality of data. Seasonal variations of targeted PAEs were analyzed using the Mann–Whitney *U* test. Differences in PAEs among residential indoor, outdoor, and personal exposure are calculated using a one-way ANOVA test. The mixed-effects model was used to calculate the within-individual (home) variance (σ^2^*_w_*) and between-individual (home) variance (σ^2^*_b_*) in personal exposure and residential indoors [37]. Statistical analyses were performed in R 3.5.1 (R Development Core Team, 2018: http://www.r-project.org accessed on: 2 July 2018). A *p*-value < 0.05 was considered statistically significant. This study applied the time-weighted exposure model to predict adult personal exposures (Appendix A).

## 3. Results

### 3.1. Characteristics of PAEs in Personal PM_2.5_


Table 1 shows the descriptive statistics of PAE congeners and ∑_6_PAEs in personal PM_2.5_ samples. In the concurrent measurement, concentration of ∑_6_PAEs in personal PM_2.5_ ranged from 0.1 to 3599.9 ng/m^3^ with an average of 699.4 ng/m^3^ (95th: 2978.4 ng/m^3^). The ∑_6_PAEs concentration accounted for 2.8 ± 3.5% of personal PM_2.5_ mass (28.9 ± 14.9 μg/m^3^). When considering the composition of PAEs in personal PM_2.5_ exposure, DEHP is the predominant congener with an average concentration of 579.6 ng/m^3^, followed by BBP (95.6 ng/m^3^), DnOP (32.9 ng/m^3^), and DnBP (20.8 ng/m^3^) (i.e., concurrent measurement; Table 1), accounting for 80.3%, 7.5%, 2.2%, and 7.9% of ∑_6_PAEs concentrations, respectively (Appendix A). 

Figure 2 illustrates the distribution of personal exposure to the studied PAEs among study participants. ∑_6_PAEs concentrations varied considerably from 13.8 to 2743.9 ng/m^3^ across individual participants. Individual PAE congeners in simultaneous personal measurements were comparable with exposures from all subjects (*p* > 0.05). Table 1 also presents the within- and between-individual variance of PAE congeners in personal PM_2.5_. Consistent with the findings for ∑_6_PAEs [38], the within-individual variance (74.5–98.9%) for individual PAE congeners accounted for a more substantial part of the total variability than the between-individual variance. A previous exposure study in Hong Kong found similar results for heavy metal(loid)s (e.g., vanadium, manganese, copper, zinc, arsenic, lead) in personal PM_2.5_, with results showing higher σ^2^*_w_* (64.5–96.9%) than σ^2^*_b_* (3.1–35.5%) from adult participants [37]. Our results demonstrated a smaller between-individual variance (1.1–25.5%), suggesting the importance of obtaining repeated samples to capture exposure variability. Similar findings were reported in previous publications, indicating balancing the relative magnitude of σ^2^*_w_* and σ^2^*_b_* to achieve an optimal measurement strategy for study participants. Chen et al. (2022) [31] discovered that repeated personal monitoring in different seasons could capture the variations in exposure concentrations of hazardous pollutants. Longer-term personal measurement is required in urban areas for accurate health risk assessment. 

The review results showed limited research for personal exposure to particle-bound PAEs. Five of the 35 included publications investigated and reported the ∑PAE concentrations in personal PM_2.5_ samples (Table 2), including one that assessed PAE congeners in different exposure categories [29,38,39,40]. For instance, He et al. (2021) [40] and Li et al. (2019) [41] investigated PM_2.5_-bound PAEs in personal exposure in rural areas of Northwest China, revealing significantly lower personal ∑_6_PAEs (e.g., 6.4 ng/m^3^, 42–54 ng/m^3^, respectively) than those measured in the current study. In contrast, the personal PM_2.5_ exposure levels were remarkably higher in Hong Kong. By contrast, Xu et al. (2019) [39] observed significantly higher personal PM_2.5_-bound PAEs (674.3–1229.4 ng/m^3^) from a group of adults (including houseworkers, students, and drivers) in Southern West Africa. The use of DEHP has been regulated because of its carcinogenic and mutagenic properties. Previous studies performed in different exposure categories revealed that DEHP was the dominant PAE congener [39,40]. In addition, Xu et al. (2019) [39] found that DEHP was the predominant PAE congener in personal PM_2.5_ exposure, with average concentrations ranging from 376.3 to 566.4 ng/m^3^.

### 3.2. Characteristics of PAEs in Residential Indoor PM_2.5_


Table 3 provides concentrations of individual PAE congeners and ∑_6_PAEs measured from residential indoor PM_2.5_. The reported ∑_6_PAEs accounted for an average of 1.8 ± 3.7% indoor PM_2.5_ level (35.1 ± 19.0 μg/m^3^). Daily residential indoor ∑_6_PAEs concentrations varied from 0.8 to 3245.4 ng/m^3^ with an average of 646.9 ng/m^3^. According to Table 3 and Appendix A, average DEHP (582.2 ng/m^3^) was presented at the highest level in residential indoor PM_2.5_, followed by BBP (65.5 ng/m^3^), DnBP (27.1 ng/m^3^), and DnOP (20.5 ng/m^3^), accounting for 80.3%, 5.8%, 11.6%, and 1.6% of ∑_6_PAEs concentrations, respectively. DMP and DEP concentrations were one to two orders of magnitude lower than other PAE congeners because these low-molecular-weight (LMW) PAEs tended to be present in the gas phase. 

Additionally, the average concentration of PAEs varied widely from one household to another (23.2–1739.8 ng/m^3^) (Appendix A). As shown in Table 2, the within-home variance (σ^2^*_w_*: 69.7%–100%) accounted for a more substantial part of the total variability for PAE congeners. There was considerable variability in between-home exposure to DMP (σ^2^*_b_*: 30.3%) and DEP (σ^2^*_b_*: 23.7%). As for DMP and DEP in residential indoors, σ^2^*_b_* was comparable with those in personal PM_2.5_ exposure in the current study. For other indoor PAE congeners, the between-home variance was less evident (σ^2^*_b_*: 0%–3.7%). The lower σ^2^*_b_* suggested that repeated measurements are needed to further explore the exposure variability. 

We compared our results with studies that measured indoor PAE exposures. As shown in Table 2, of the 35 included studies, 13 investigated PM_2.5_-bound PAE concentrations in different indoor settings (e.g., residential indoors, office, school, and dormitory). Most of these studies were performed in Chinese cities, two in European cities, and one in American cities [22,44,46]. The average indoor PM_2.5_-bound ∑_6_PAEs concentrations in the current study were considerably lower than those reported in Beijing (office: 856.6–1288.0 ng/m^3^; residential indoor: 2299.0–4436.8 ng/m^3^) [27,42] and Hangzhou, China (office: 1375.8 ng/m^3^; newly decorated residence/office: 1576.7 ng/m^3^) [45,48], but were elevated when compared with rural areas (e.g., households in Northwest China: 60.9–75.8 ng/m^3^) [40]. In addition, comparable or lower indoor PM_2.5_-bound PAEs concentrations were reported in several other Chinese cities (e.g., Xi’an (school: 788.9 ng/m^3^) [47] and Tianjin, China (residence indoor: 179.2 ng/m^3^) [43]. However, the corresponding indoor PM_2.5_ concentrations in these cities were substantially higher when compared with Hong Kong. 

Residential indoor DMP and DEP concentrations were significantly lower than those measured in different study areas (e.g., urban and rural areas) (Table 2). Four PAE congeners (including DnBP, BBP, DEHP, and DnOP) were on the European Community priority pollutants blacklist [49]. Globally, the measured PM_2.5_-bound DEHP concentrations measured in this investigation were more severe compared with previous results from indoor air in most Chinese cities and developed countries (Table 2), such as the U.S. (56–79 ng/m^3^) and Norway (12 ng/m^3^) [44,46]. Overall, residential indoor and personal PAE exposure showed similar composition profiles, and DEHP has made the dominant contribution to ∑_6_PAEs, followed by BBP, DnBP, and DnOP (Appendix A). However, DnBP showed a relatively higher fraction in residential indoors than in personal exposure. Additionally, the magnitude and composition characteristics of PAE in residential indoors are consistent with those in urban indoor microenvironments (e.g., residences, offices, classrooms) in several Chinese cities, which exhibited a similar pattern showing that PM_2.5_-bound DEHP was the most abundant PAE congener [45,47,48]. Notably, some studies demonstrated different patterns [44]. For example, some studies performed in Beijing showed that DnBP was the predominant particle-bound PAE congener in schools and residential indoors [35,42,50]. 

### 3.3. Residential Indoor/Outdoor and Personal Exposure/Outdoor Relationships 

The average concentration of outdoor ∑_6_PAEs during the entire sampling period ranged from 183.3 to 3814.1 ng/m^3^ (Appendix A). No significant spatial variations were shown for PAEs across these three sampling sites (Appendix A). Appendix A compares PAE concentrations in outdoor PM_2.5_ in different study areas. The average ∑_6_PAEs concentrations measured in ambient (outdoor) air in Chinese cities were remarkably lower than the current results (4.2–834.3 ng/m^3^). A review article corroborated these findings, and the results indicated an increasing trend in population exposure to DnBP, BBP, and DEHP from 2001 onwards in China [51]. When considering the individual PAE congeners, DEHP is the most abundant PAEs in ambient PM_2.5_ in Hong Kong, with an average concentration of 1125 ng/m^3^. The current study found that outdoor DEHP concentrations were much higher than those reported in other Chinese cities (1.2–875.8 ng/m^3^). In addition, the average concentration for BBP and DEHP in ambient PM_2.5_ (Appendix A) exhibited a dramatic increase (~90%) compared with those in 2003 (n.d.–0.8 ng/m^3^ and 96–241 ng/m^3^, respectively) in Hong Kong [20]. PAEs account for ~90% of plasticizer and PVC production, and Guangdong Province had one of the highest production rates due to the high industrial demands. Eales et al. (2022) [6] indicated that anthropogenic activities (e.g., long-range transport from PAE manufacturing, distribution, and discharge) influence PAE distribution in the atmosphere, resulting in high concentrations in the urban area. The wide application of plastic products could also contribute to the increase in PAE concentrations. Therefore, more attention should be paid to the variation of PAE profiles and the higher exposure levels of PAEs in highly urbanized areas. 

This study obtained concurrent outdoor, residential indoor, and personal PAEs from 26 households. As shown in Table 3, comparing the average concentrations of PAEs indoors and outdoors, the average I/O ratios of PAE congeners and ∑_6_PAEs ranged from 1.8 to 4.8. For paired data, Appendix A shows that indoor and personal ∑_6_PAEs exceeded the corresponding outdoor levels in 33.3%–50.0% of participants or households; the median I/O ratios for PAE congeners < 1. As for individual PAE congeners, the highest average I/O ratio was shown for DnBP (4.8), and there were significant differences for outdoor with indoor and personal DnBP exposure (*p* = 0.02), suggesting that DnBP sources are primary in some residential indoors. A possible reason for significant differences could be the combined effects of outdoor and indoor origin exposure. Otake et al. (2004) [23] found elevated DnBP concentrations (6.18 μg/m^3^) in Japan’s newly built residential houses, indicating that vinyl cloth or paint may be the main sources. A study performed in a primary school in Barcelona, Spain, revealed that the higher particle-bound DnBP observed in the school could be attributed to the older classroom materials [22]. Studies on particle-bound PAE I/O ratios are scarce. Recent studies in Beijing [35] and Hangzhou [48] found that PAE I/O ratios in the particle phase were 3.2–5.0 and 3.5–5.9 in different indoor settings. Previous results showed that the DEHP concentration in indoor PM_2.5_ was 1.4–1.9 times higher than in outdoor [35,47]. The distinct patterns were associated with the physical properties of PAE congeners. For example, as temperature increases, the gas/particle partitioning of PAE favors the gas phase. Further, DMP and DEP concentrations in different exposure categories demonstrated no significant differences but higher average and median I/O ratios compared with other PAE congeners. The result was consistent with previous findings; for example, He et al. (2019) [52] reported higher fractions of DMP and DEP indoors than in outdoor air. 

Seven of the eight studies that reported paired indoor–outdoor PAE concentrations (Table 2) were conducted in urban areas with intensive anthropogenic activities. Only one was performed in a rural area [41]. In the current study, a distinct pattern for particle-bound ∑_6_PAEs concentration was discovered (outdoor (1101.5 ng/m^3^) > indoor (646.9 ng/m^3^)). Wang et al. (2019) [50] found that concentrations of outdoor PAEs (1549.0 ng/m^3^) were higher than those measured indoors (940.2 ng/m^3^) during the haze periods in Beijing, China. Studies also suggested that high outdoor PAEs may be attributable to indoor sources in densely populated areas [21]. By contrast, most studies reported higher average indoor PAE concentrations compared with outdoor/ambient PAE concentrations (Table 2), suggesting the denser presence of PAE sources in some residential indoors [23]. For example, ∑_6_PAEs concentrations in some residential indoors (e.g., Subject ID 56) were higher than the concurrent outdoors (Appendix A), with the results showing I/O ratios > 5 for DnBP, BBP, DEHP, and DnOP. Additionally, residential locations for Subjects 24 and 55 were in the vicinity of landfill sites in Hong Kong. Previous studies have shown that people living near hazardous waste disposal sites and landfills might experience exposure to higher-than-average levels of PAEs (e.g., DEHP) from ambient air [53]. These discrepancies in indoor and outdoor variations could also be related to the temperature and seasonal differences, building ages, decoration, and the ratio of gas/particle-phase PAEs indoors and outdoors [48]. 

The current study shows slightly different seasonal patterns for PAEs in personal exposure, residential indoor, and outdoor (Appendix A). For example, individual PAE congeners in personal PM_2.5_ demonstrated higher levels in winter than in summer (*p* > 0.05). Similar seasonal patterns were shown for PAEs in residential indoors (winter > summer). Indoor air conditioning was widely used in the summer season by Hong Kong people. These results are consistent with findings in other Chinese cities [42], suggesting PAEs partitioning to particles indoors at lower temperatures [54]. However, the outdoor PAEs demonstrated an opposite seasonal pattern. Higher concentrations were observed in summer than in winter despite no significant fluctuations. The results were consistent with previous findings [55]. For example, PAEs in ambient PM_2.5_ in Hong Kong [20] and Guangzhou, China [56] showed a seasonal pattern of summer > winter. Seasonal variations for PAEs in ambient air could be attributed to emission sources (industry, anthropogenic activities) and dispersion conditions [52]. Previous investigations have suggested that the higher outdoor PAE concentrations in summer could result from the enhanced volatilization of plastics under high-temperature conditions, followed by deposition of the pre-existing particles in the ambient atmosphere [57]. 

### 3.4. Source Identification 

Table 4 presents Spearman’s correlations separately for individual PAE congeners in each exposure category. Significant correlations were shown between DMP and DEP (*r*_s_ = 0.72–0.74, *p* < 0.01). In addition, strong correlations for DEHP with DnBP (*r*_s_ = 0.81–0.88; *p* < 0.01), BBP (*r*_s_ = 0.83–0.87; *p* < 0.01), and DnOP (*r*_s_ = 0.87–0.90; *p* < 0.01) were shown in personal exposure and residential indoor. Similarly, strong correlations were demonstrated outdoors. There are moderate correlations between DnBP with DMP and DEP in personal exposure, suggesting that there might be common sources for these compounds. No such associations were found outdoors. Outdoor monitoring at fixed sites could not capture indoor origin pollutants. Net et al. (2015) [5] found that half-lives of PAEs indoors are longer than outdoors. The current results showed no clear association for individual PAE congeners among different exposure categories, suggesting that sources contributing to PAE exposure differed in each exposure category. Previous findings demonstrated significant discrepancies for ∑_6_PAEs among residential indoor, outdoor, and personal PM_2.5_ exposure [38], with results showing a coefficient of divergences > 0.70. The results showed weak correlations between ∑_6_PAEs and PM_2.5_ concentrations in different exposure categories (e.g., personal exposure (*r* = 0.07), residential indoor (*r* = 0.21), outdoor (*r* = 0.14)); weak associations were found for ∑_6_PAEs with organic carbon (*r* = 0.18), elemental carbon (*r* = 0.09), and ∑parent-PAHs (*r* = 0.19) [29,38]. 

We applied PCA to explore the sources of particle-bound PAEs in PM_2.5_. Table 5 describes the factor loadings (rotated component matrix) from PCA in personal, residential indoor, and outdoor PAEs. For personal PAE exposure, two principal components were identified, which explained 74.9% of the total variance. Component 1 was characterized by high loadings of BBP (0.91), DEHP (0.89), DnOP (0.83), and a lesser extent of DnBP (0.50). DEHP and DnOP have been used in household products. Aylward et al. (2009) [58] reported that DEHP is widely used as a plasticizer for PVC and construction materials. BBP is commonly used as a plasticizer for PVC flooring. Studies have indicated that tire wear could be an outdoor source for indoor DnBP. In this study, Component 1 can be linked to plastics from household products and building materials and the infiltration of outdoor emissions. Component 2 accounted for 31.9% of the total variance and comprised DMP (0.93), DEP (0.87), and DnBP to a lesser degree. These LMW PAEs (e.g., DMP, DEP, DnBP) are extensively used as additives in cosmetics and household and personal care products [59,60]. 

For residential indoor PAEs, three principal components accounted for 88.3% of the total variance. Component 1 explained 33.5% of the total variance and comprised BBP (0.90), DnOP (0.93), and a lesser extent of DEHP (0.58), indicating the influence of widely used plasticizers in PVC and other polymer products. Component 2 in residential indoors was loaded with DnBP (0.96) and DEHP (0.76), which explained 28.0% of the total variance. A statistical survey revealed that the current plasticizer-related sector in China is based primarily on DnBP and DEHP. The analyzed PAEs have been extensively used in household products and plasticizers. PAEs indoors could be attributable to the continuous release from nonplastic sources and those containing plastics (e.g., plastics and consumer goods). Wang et al. (2017) [61] found significantly higher concentrations of DnBP and DEHP in house dust. Component 2 could be related to fine particles generated indoors due to continuous resuspension of dust and products containing plastic. Component 3 accounted for 26.8% of the data variance and had high loading of DMP and DEP, indicating the household’s non-plastic sources (cosmetics, perfumes, and personal care products). The indoor sources of PAEs are more diverse and complicated. It is difficult to disentangle these sources because of a lack of specific observations concerning factors influencing indoor PAE exposure (e.g., plastic products, wall coverings, furniture, and building characteristics). The differences in source profiles between personal exposure and residential indoors were possibly caused by variations of indoor sources (office, home) and individual activities. However, such information is scarce, and the research gap should be filled in future studies. 

As for outdoors, two factors explained 75.7% of the data variance in this study. Factor 1 was loaded with DnBP, BBP, DEHP, and DnOP, which explained 50.0% of the total variance. As presented in Table 5, Component 1 in personal exposure (characterized by high loadings of BBP, DEHP, DnOP, and DnBP) closely resembled the corresponding sources profile for factor 1 in outdoor air. Studies indicated that outdoor DnBP and DEHP were mainly from rubber and plastic production, fuel emissions, and end-use commercial PVC products [6]. In addition, Huang et al. (2022) [56] found that industrial activities and municipal waste treatment facilities (e.g., e-waste dismantling) could be the sources of PM_2.5_-bound DEHP in the ambient air of Guangzhou. Component 1 could also be related to PAEs released from sludges and the surrounding environment into the atmosphere. Component 2 outdoors was characterized by high DMP and DEP load, contributing 25.7% of the total variance. Strong correlations were found between outdoor DMP and DEP (0.68, *p* < 0.01), and factor 2 can be linked to sources related to non-PVC products (e.g., surface coating materials). 

A new restriction set out in Commission Regulation 2018/2005 was enacted on 7 January 2019 in Hong Kong (https://research.hktdc.com/en/article/NDc5NjE0MDcw, accessed on 9 July 2020) to restrict four PAEs (DEHP, DBP, BBP, diisobutyl phthalate) in consumer goods placed on the market. In this investigation, we further assessed the health risks of exposure to major PAE congeners (e.g., DnBP, BBP, DEHP, and DnOP). 

### 3.5. Variations in Exposure Intake of PAEs via Inhalation

Daily intake of PAEs varied by participants’ age, weight, inhalation rate, and exposure concentrations in individual participants (Equation (2)). The age of the study participants ranged from 18 to 42 years, and the average weights for males and females were 65.2 and 55.2 kg, respectively. The average inhalation rates for male and female participants were 15.4 ± 2.1 and 14.9 ± 2.1 m^3^/day (Appendix A). The inhalation rates were consistent with those in the U.S. EPA’s *Exposure Factors Handbook*, which reported inhalation rates of 14.7–14.9 m^3^/day for male and female adults [32]. Time activities for the study participants are reported in Appendix A. The time-weighted exposure concentrations for PAEs were assessed based on the above parameters by incorporating outdoor and residential indoor PAE concentrations with individual activity patterns (Appendix A). According to Appendix A, LMW PAEs and high-molecular-weight PAEs demonstrated different patterns. The estimated exposure concentrations for LWM PAEs (i.e., DMP, DEP) were significantly higher than the measured concentrations. These results suggest that both gas- and particle-phase PAEs should be considered when conducting personal measurements. PAE exposure levels in male and female participants were similar, and no apparent gender variation in composition profiles was shown (Appendix A). 

Table 6 reports the daily intake of exposure to PAE congeners via the inhalation route based on measured and estimated personal exposure. Average daily intakes (DI*_inh_*) of the targeted PAEs illustrated the order of DEHP (0.12 μg/kg-day) > BBP (0.018 μg/kg-day) > DnOP (0.006 μg/kg-day) > DnBP (0.005 μg/kg-day). Similarly, if we consider the estimated PAE exposures, the daily intake of PAEs followed the order of DEHP (0.16 μg/kg-day) > BBP (0.015 μg/kg-day) > DnBP (0.005 μg/kg-day) > DnOP (0.006 μg/kg-day). The average daily intake of particle-bound DMP and DEP was over three orders of magnitude lower than DEHP. A significant difference in the daily intake of PAEs was shown for DnOP; this may result from the fact that higher outdoor and indoor DnOP concentrations contribute to personal exposure (Appendix A). Our results showed a lower level of exposure, generally lower than those in Northern Chinese cities. Bu et al. (2019) [62] found that the average DEHP intakes were lower than those modeled for DEHP inhalation exposure for Chinese adults (0.35–0.31 μg/kg-day). The reference dose (RfD) for PAE congeners was adapted from the U.S. EPA. RfDs of 20 μg/kg-day for DEHP, 200 μg/kg-day for BBP, 100 μg/kg-day for DnBP, and 400 μg/kg-day for DnOP have been suggested in the Integrated Risk Information System (IRIS) [63]. In this investigation, the average and 95th daily intake of inhalation exposure to BBP, DnBP, and DnOP were lower than the recommended RfDs. 

Table 6 shows that the measured and estimated 95th daily intake values were 0.43–0.50 μg/kg-day. He et al. (2019) [52] found that DEHP was the most abundant congener in the atmosphere compared with other environmental media. Considering the 95th DEHP exposure concentration outdoors (3369.6 ng/m^3^) and residential indoors (1883.0 ng/m^3^), the estimated daily intake will be 0.69 μg/kg-day. No noticeable gender differences were shown in the exposure concentrations and daily inhalation intake of PAEs between male and female subjects (Appendix A). The results concerning the intake of DEHP in different exposure categories were lower than the RfD of 20 μg/kg-day recommended by the U.S. EPA, but this does not mean that there are no harmful effects of PAE exposures. We must consider that daily intake in the current study is calculated based on particle-phase DEHP exposure, and gas-phase PAEs would also exert toxic effects on human health. Zhang et al. (2019) [55] found that the average DMP, DEP, DnBP, and DEHP concentrations in the gas phase were 3.5–12 times higher than the particle-phase concentrations. In addition, previous studies indicated that inhalation exposure to PAEs varied with age groups [62]. For example, Chen et al. (2018) [35] demonstrated a higher intake of DEHP in infants than in students and office workers. Additionally, Wang et al. (2019) [27] reported that the inhalation doses of PAEs for infants (0.845 μg/kg-day) and children (0.203 μg/kg-day) were 3–12 times higher than that in adults (0.07 μg/kg-day). 

### 3.6. Health Risk Assessment of Inhalation Exposure to Targeted PAE Congeners 

Different health effects were attributable to specific components of PM_2.5_ [19]. To put measured and estimated values in the context of related health risks, we assessed inhalation risks of exposure to particle-bound DEHP due to its carcinogenetic potential. Huang et al. (2022) [56] suggested that body weight is a critical factor influencing the cancer risk estimation for inhalation exposure to DEHP, followed by inhalation rate and exposure concentrations. Thus, we incorporated these variables in the assessment at the individual level. As shown in Table 6, the average cancer risks attributable to DEHP inhalation exposure from estimated exposure (CR*_inh_estimated_*) and directed measured exposure (CR*_inh_*) for adults were 1.2 × 10^−6^ and 1.3 × 10^−6^, slightly higher than the acceptable risk level (1.0 × 10^−6^) recommended by the U.S. EPA. The direct measured personal exposure reflects the possible health risks posed by particle-bound DEHP for adults from indoors and outdoors. 

We employed the Monte Carlo simulation to investigate the distribution of risks attributable to DEHP inhalation exposure in adults. Figure 3 shows the probability distribution of CR*_inh_* caused by inhalation exposure to DEHP for adults. The mean and 95th CR*_inh_* attributable to DEHP inhalation exposure were 1.3 × 10^−6^ and 4.6 × 10^−6^. In addition, the mean and 95th cancer risks resulting from the estimated DEHP exposure were 1.2 × 10^−6^ and 3.5 × 10^−6^ (Figure 3). In this investigation, cancer risk estimates were similar for male and female adults (*p* > 0.05). Consistent results about the negative impacts of PAE exposure in ambient PM_2.5_ were demonstrated in other regions/countries [64,65]. Wang et al. (2019) [50] reported that that carcinogenic risk attributed to inhalation exposure to indoor DEHP was 4.1 × 10^–6^ and 10.5 × 10^–6^ for adults and children, respectively, indicating that indoor DEHP exposure might adversely impact the adult populations in Beijing, China. Previous results revealed a lower cancer risk from DEHP exposure in adults than in children, and cancer risks decreased with age increment. Children in the same household could have higher cancer risks attributed to DEHP exposure. Wang et al. (2019) [27] found that higher DEHP exposure risks in children could be attributable to higher inhalation rates and more frequent hand-to-mouth activities. 

The DEHP exposure concentrations and CSF value variability could result in uncertainty in health risk estimations. For example, if we consider the CSF value of 0.014 (mg/kg·day)^−1^ from the IRIS database, the potential cancer risks would be one order of magnitude higher (i.e., 2.0 × 10^−5^ and 2.1 × 10^−5^) than the acceptable risk level. Our results exemplify the potential carcinogenic effects of DEHP on human health, suggesting that long-term personal DEHP exposure warrants extensive investigation and possibly remediation.

In this work, some limitations need to be considered. First, subjects’ participation availability limits the sample size, and repeated sampling from a larger population is required. The short-term PAE exposure does not apply to all subpopulations or other study areas. Second, there were insufficient data on “other” microenvironments (e.g., public places) and indoor factors (e.g., construction, decoration, household items) that may potentially affect residential PAE levels. When interpreting the study results, considering uncertainties in inhalation exposure estimation is required. Additional exploration concerning characteristics of the particle- and gas-phase PAE congeners in different exposure metrics may provide information about exposure errors introduced by sampling methods. Despite these limitations, our results reinforce the hypothesis that particle-bound PAE exposure poses potential cancer risks to the adult population in Hong Kong. 

## 4. Conclusions

This investigation revealed a comprehensive picture of the abundance and composition of PAE congeners in personal PM_2.5_ exposure, residential indoor, and outdoor. The within-individual and within-home variances dominated the total variability of PAE congeners in personal and individual indoors. DEHP was the dominant PAE congener, contributing to 80.4%–85.0% of ∑_6_PAEs, followed by BBP, DnBP, and DnOP. DEHP concentrations were significantly higher in Hong Kong compared with other studies. Significant correlations were observed for DEHP with BBP, DnBP, and DnOP and DMP with DEP, suggesting their coexistence in each exposure category. However, the results showed strong heterogeneity for PAE congeners, and no apparent intercorrelations were observed across different exposure categories. We further explored the emission sources and inhalation health risks of exposure to PAEs. A time-weighted exposure model was used to estimate PAE exposures. Daily intakes of inhalation exposure for DEHP, BBP, DnBP, and DnOP were lower than U.S. EPA limits. Nevertheless, the assessment for inhalation exposure to DEHP poses potential cancer risks to human health, and CR*_inh_* from DEHP exposure exceeded the U.S. EPA threshold of 1.0 ×10^–6^ for adults. Our results raised concerns over the elevated inhalation risk of exposure to particle-bound DEHP for Hong Kong residents. These results can help facilitate future regulatory action and evidence-based strategies to reduce PAE exposures and protect vulnerable subpopulations (e.g., infants and young children). 

## Figures and Tables

**Figure 1 ijerph-19-13425-f001:**
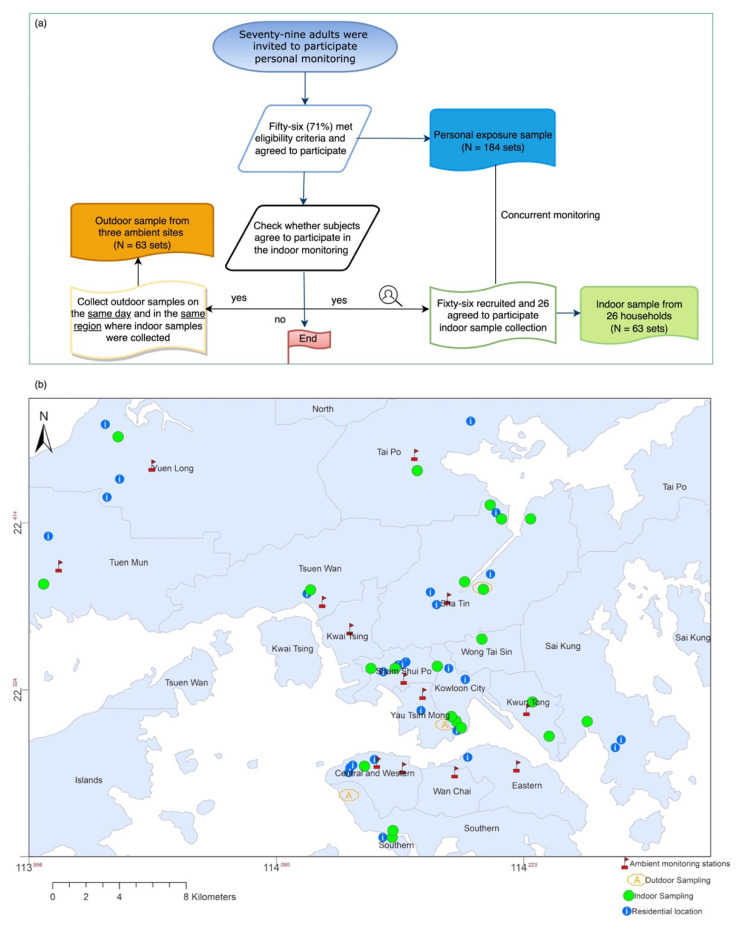
(**a**) Flowchart of the study design and (**b**) map showing the residential location of individual participants (i.e., personal exposure: blue) and sampling households (i.e., residential indoor: green) in this study.

**Figure 2 ijerph-19-13425-f002:**
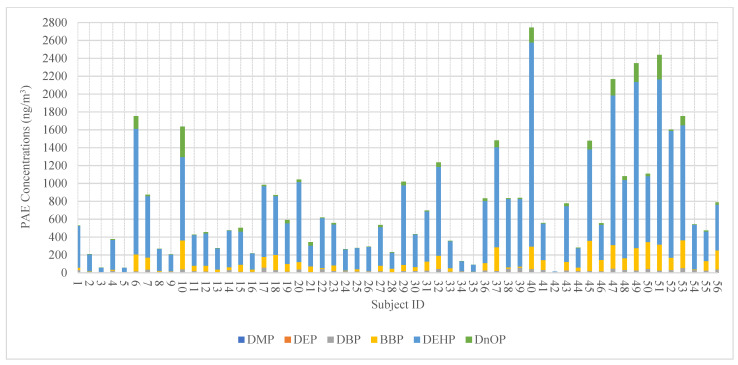
The ∑_6_PAEs (the sum of six PAE congeners, including DMP, DEP, DnBP, BBP, DEHP, and DnOP) concentrations and PAE congener profiles (ng/m^3^) that quantified in personal PM_2.5_ in individual participants.

**Figure 3 ijerph-19-13425-f003:**
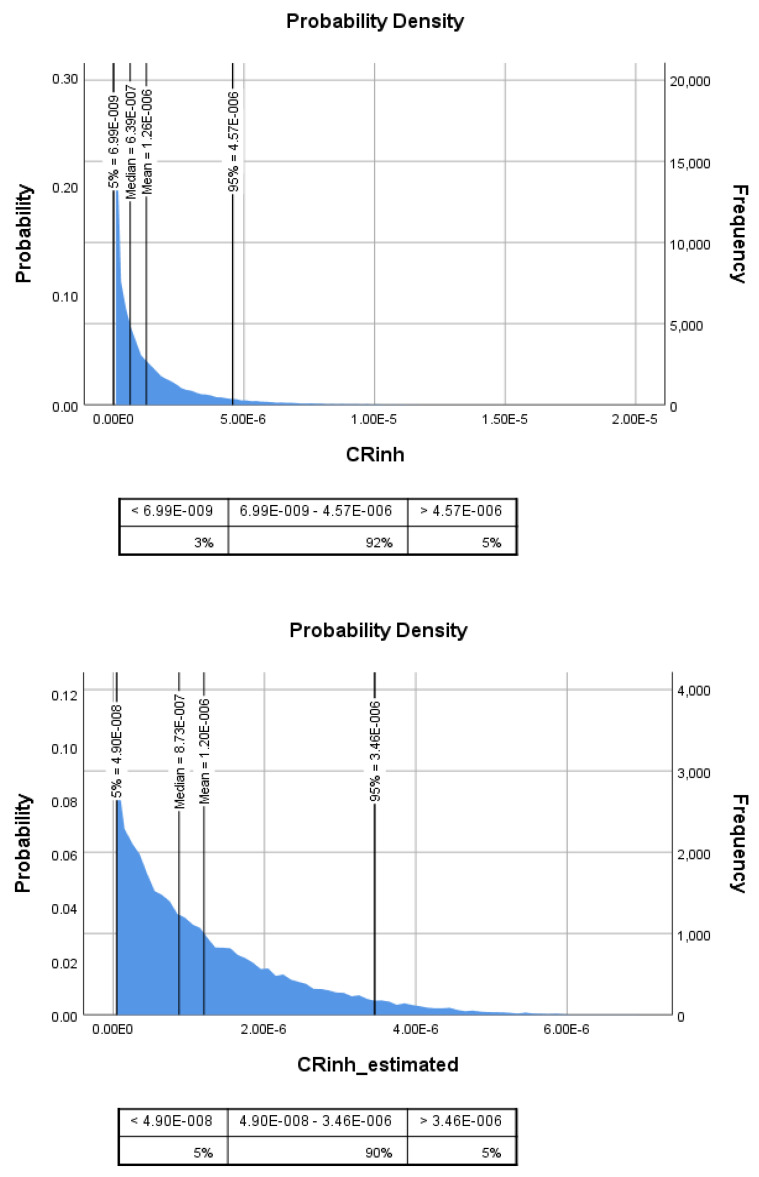
Inhalation cancer risk resulting from personal exposure to DEHP (CR*_inh_*) and estimated exposure to DEHP (CR*_inh_estimated_*) using the time-weighted model. Note: 1.0E-06 = 1.0 × 10^−6^.

**Table 1 ijerph-19-13425-t001:** Summary statistics of individual PAE congeners and ∑_6_PAEs (ng/m^3^) in personal PM_2.5_ samples from study participants.

ng/m^3^	Concurrent					Total							Total vs. Concurrent
	Mean ± SD ^a^	Median	95th ^g^	Min–Max ^b^	N ^c^	Mean ± SD	Median	95th ^g^	Min–Max	N	σ^2^*_b_* (%) ^d^	σ^2^*_w_* (%) ^e^	*p*-Value ^f^
DMP	0.10 ± 0.07	0.08	0.28	0.00–0.32	56	0.11 ± 0.09	0.08	0.23	0.003–0.61	170	24.5	75.5	0.62
DEP	2.10 ± 2.49	1.58	6.52	0.01–15.85	58	2.13 ± 2.25	1.57	6.22	0.01–15.85	176	22.3	77.7	0.94
DnBP	20.8 ± 20.7	13.8	72.9	0.02–81.6	59	21.7 ± 21.8	14.8	63.1	0.02–107.0	180	1.1	98.9	0.76
BBP	95.6 ± 153.0	7.9	420.1	0.03–598.1	54	93.4 ± 143.4	11.1	394.7	0.02–678.3	165	11.5	88.5	0.93
DEHP	579.6 ±719.2	390.3	2201.6	0.27–3002.3	57	605.7 ± 786.1	361.4	2433.0	0.16–5134.5	175	14.5	85.5	0.82
DnOP	32.9 ± 70.1	6.7	211.4	0.13–353.7	52	40.1 ± 104.2	6.3	195.7	0.04–929.9	153	25.5	74.5	0.58
∑_6_PAEs ^h^	699.4 ± 899.9	364.3	2978.4	0.10–3599.9	59	732.4 ± 970.2	409.7	3057.1	0.10–6146.7	180	11.0	89.0	0.81

Notes: ^a^ SD: standard deviation. ^b^ Min–Max: minimum–maximum. ^c^ N refers to the number of valid data, and concentrations below the method detection limits (MDLs in Appendix A) were discarded. ^d^ σ^2^_b_: between-individual variance (%). ^e^ σ^2^_w_: within-individual variance (%). ^f^ The level of significance was taken as *p*-value < 0.05. ^g^ 95th: The 95th percentile. ^h^ ∑6PAEs: the sum of six PAE congeners, including DMP, DEP, DnBP, BBP, DEHP, and DnOP.

**Table 2 ijerph-19-13425-t002:** Global comparison for particle-phase PAEs in residential indoor (along with concurrent outdoor) and (or) personal PM_2.5_ exposure (unit: ng/m^3^). Mean values of each PAE congener were reported.

Exposure Category	Site (Study Area)	Season	DMP	DEP	DnBP	BBP	DEHP	DnOP	Mean (SD)	Range (Min–Max)	References
Indoor (residential)	Beijing, China	Summer	39.5	21.1	1879.1	11.6	341.9	5.8	2299.0	nr	Huang et al. (2020) [42]
			55.4	8.0	2973.2	2.2	1304.9	93.1	4436.8	nr	
Indoor (residential)	Tianjin, China	Mixed	2.99	0.75	130.7	0.32	44.4	0.099	179.2	7.3–1244	Zhang et al. (2014) [43]
Indoor	Norwegian, Norway	nr	^—^	^—^	74	11	12	^—^	97	na	Rakkestad et al. (2007) [44]
Indoor (office)	Hangzhou, China	Mixed	223.6	269.7	321.2	153	408.3	nd	1375.8	774–2050	Song et al. (2015) [45]
Indoor (office)	Beijing, China	Spring	5.2	1.6	936.2	0.4	344	0.6	1288.0	na	Wang et al. (2019) [27]
	Beijing, China	Winter	44.7	2.7	529.8	1.1	277.6	0.8	856.6	na	
Outdoor	Beijing, China	Spring	52.3	2.6	179.7	3.1	217.4	0.7	455.8	na	
	Beijing, China	Winter	203.7	3.2	363.3	0.1	244.4	0.6	815.2	na	
Outdoor	Northern California, US	Mixed	^—^	^—^	^—^	^—^	17	^—^	na	na	Brody et al. (2009) [46]
	Northern California, US	Mixed	^—^	^—^	^—^	^—^	<MDL	^—^	na	na	
Indoor	Northern California, US	Mixed	^—^	^—^	^—^	^—^	79	^—^	na	na	
	Northern California, US	Mixed	^—^	^—^	^—^	^—^	56	^—^	na	na	
Indoor (school)	Barcelona, Spain	nr	^—^	^—^	270	^—^	^—^	^—^	na	na	Van Drooge et al. (2020) [22]
Outdoor	Barcelona, Spain	nr	^—^	^—^	12	^—^	^—^	^—^	na	na	
Indoor (school)	Barcelona, Spain	nr	^—^	^—^	92	^—^	^—^	^—^	na	na	
Outdoor	Barcelona, Spain	nr	^—^	^—^	9	^—^	^—^	^—^	na	na	
Indoor (school)	Xi’an, China	May	32.45	68.15	282	43	328.2	35.15	788.9	nr	Wang et al. (2017) [47]
Outdoor	Xi’an, China	May	34.5	41.4	186.9	41.4	236.4	33.8	574.4	nr	
Indoor (dormitory)	Beijing, China	Mixed	4.33	7.02	368	7.36	65	15.9	468	nr	Chen et al. (2018) [35]
Indoor (office)	Beijing, China	Mixed	5.92	6.16	118	10.9	123	16.3	280	nr	
Indoor (residential)	Beijing, China	Mixed	3.54	7.38	350	41.2	84.1	11.9	498	nr	
Outdoor	Beijing, China	Mixed	6.15	6.8	43.5	10.1	39	19.3	125	nr	
Indoor (newly decorated residence/ordinary/office)	Hangzhou, China	Mixed	135.1	207.1	190.43	222.9	522.6	298.6	1576.7	nr	Ouyang et al. (2019) [48]
Outdoor	Hangzhou, China	Mixed	15.6	32.9	40	45.4	104	82.9	320.8	nr	
Personal	Hong Kong, China	Summer	^++^	^++^	^++^	^++^	^++^	^++^	471	0.1–3800	Fan et al. (2018) [29]
Personal	Abidjan, West Africa	Mixed	2.2	8.3	224.8	13.8	566.4	40.9	856.4	nr	Xu et al. (2019) [39]
Personal	Abidjan, West Africa	Mixed	9.6	146.5	440.7	248.2	8.1	376.3	1229.4	nr	
Personal	Cotonou, West Africa	Mixed	1.9	6.8	248.2	8.1	376.3	33	674.3	nr	
Outdoor, indoor, personal	Hong Kong	Mixed	^++^	^++^	^++^	^++^	^++^	^++^	606	0.2–5135	Chen et al. (2020) [38]
Indoor	Rural Xian, China	Winter	<MDL	<MDL	2.4	^—^	2.1	bd	na	na	Li et al. (2019) [41]
Outdoor		Winter	<MDL	<MDL	2	^—^	1.8	bd	na	na	
Personal		Winter	<MDL	<MDL	2.8	^—^	3.3	0.12	na	na	
Indoor ^a^	Rural Xian, China	Winter	4.9	1.7	0.57	1.6	25	42	75.8	nr	He et al. (2021) [40]
Indoor ^b^	Rural Xian, China	Winter	16	4.4	0.57	3.9	16	20	60.9	nr	
Personal ^a^	Rural Xian, China	Winter	0	0.77	0.82	0.59	35	59	96.2	nr	
Personal ^b^	Rural Xian, China	Winter	0.02	1.1	0.76	0.99	29	32	63.9	nr	

Notes: ^++^: analyzed, but no specific values were reported. ^—^: not analyzed. nr: not reported. nd: not detected; <MDL: blow method detection limit; mean: average value of summation of six PAE congeners; SD: standard deviation; range: minimum–maximum; ^a^ using coal as domestic fuel; ^b^ using coal and biomass as domestic fuel.

**Table 3 ijerph-19-13425-t003:** Summary statistics of PAE congeners and ∑_6_PAEs in residential indoor PM_2.5_.

	Indoor (ng/m^3^)							I/O Ratio (No Unit)			
	Mean ± SD ^a^	Median	95th ^h^	Min–Max ^b^	N ^c^	σ^2^*_b_* (%) ^e^	σ^2^*_w_* (%) ^f^	Mean ± SD	Median	Q1–Q3 ^g^	N ^d^
DMP	0.17 ± 0.17	0.12	0.43	0.01–1.08	61	30.3	69.7	1.9 ± 4.3	0.81	0.52–1.46	57
DEP	3.22 ± 3.07	2.21	9.25	0.05–16.11	62	23.7	76.3	2.3 ± 5.8	0.84	0.36–1.75	60
DnBP	27.1 ± 24.9	21.2	71.10	0.1–129.0	63	0	100	4.8 ± 15.2	0.69	0.34–1.93	57
BBP	65.5 ± 122.5	12.9	315.6	0.2–654.4	53	0	100	1.9 ± 3.5	0.27	0.03–1.71	41
DEHP	582.2 ± 604.8	409.5	2007.4	0.4–2330.6	59	3.7	96.3	1.8 ± 3.3	0.46	0.16–1.75	50
DnOP	20.5 ± 50.1	6.4	98.9	0.1–243.0	50	1.6	98.4	2.8 ± 7.5	0.45	0.06–3.04	40
∑_6_PAEs ^i^	646.9 ± 734.1	471.8	2495.5	0.8–3245.4	63	7.2	92.8	3.2 ± 11.6	0.46	0.16–1.76	55

Notes: ^a^ SD: standard deviation. ^b^ Min–Max: minimum–maximum. ^c^ N refers to the number of valid data, and concentrations below the method detection limits (MDLs in Appendix A) were discarded. ^d^ Outliers that fell outside the upper outer fence (i.e., Q3 + 3*(O3–Q1)) (e.g., I/O > 100) were discarded. ^e^ σ^2^*_b_*: between-home variance (%). ^f^ σ^2^*_w_*: within-home variance (%). ^g^ Q1–Q3: the first quartile–the third quartile. ^h^ 95th: the 95th percentile. ^i^ ∑_6_PAEs: the sum of six PAE congeners, including DMP, DEP, DnBP, BBP, DEHP, and DnOP.

**Table 4 ijerph-19-13425-t004:** Spearman’s correlation matrix for PM_2.5_-bound PAE congeners in (and between) personal exposure, residential indoor, and outdoor, respectively.

Personal Exposure (P)	DMP	DEP	DnBP	BBP	DEHP	DnOP
DMP	1	0.74 **	0.51 **	−0.05	0.24 **	0.005
DEP		1	0.30 **	−0.21	0.09	−0.16
DnBP			1	0.63 **	0.81 **	0.58 **
BBP				1	0.87 **	0.90 **
DEHP					1	0.90 **
DnOP						1
Residential indoor (I)						
DMP	1	0.72 **	0.30 *	−0.003	0.30 *	0.05
DEP		1	0.07	−0.16	0.03	−0.04
DnBP			1	0.73 **	0.88 **	0.76 **
BBP				1	0.83 **	0.80 **
DEHP					1	0.87 **
DnOP						1
Outdoor (O)						
DMP	1	0.68 **	0.09	0.11	0.11	0.07
DEP		1	−0.0005	−0.16	−0.11	−0.21
DnBP			1	0.77 **	0.90 **	0.81 **
BBP				1	0.90 **	0.93 **
DEHP					1	0.93 **
DnOP						1
P-O	−0.05	−0.12	−0.10	0.14	−0.14	−0.03
I-O	0.14	0.02	0.03	−0.15	−0.13	−0.16
P-I	0.23	0.17	−0.18	0.07	−0.03	−0.02

Notes: ** *p* < 0.01. * *p* < 0.05.

**Table 5 ijerph-19-13425-t005:** Factor loading of principal component analysis (PCA) on PAEs in personal exposure, residential indoor, and outdoor PM_2.5_. Varimax with Kaiser normalization.

	Personal Exposure		Residential Indoor			Outdoor	
Species	PC ^a^ 1	PC2	PC1	PC2	PC3	PC1	PC2
DMP	^b^	0.93	^b^	^b^	0.86	^b^	0.87
DEP	^b^	0.87	^b^	^b^	0.91	^b^	0.87
DnBP	0.50	0.52	^b^	0.96	^b^	0.74	^b^
BBP	0.91	^b^	0.90	^b^	^b^	0.90	^b^
DEHP	0.89	^b^	0.58	0.76	^b^	0.93	^b^
DnOP	0.83	^b^	0.93	^b^	^b^	0.87	^b^
Eigenvalue	2.58	1.91	2.67	1.61	1.00	3.00	1.54
% of variance ^c^	43.0	31.9	33.5	28.0	26.8	50.0	25.7

Notes: ^a^ PC: principal component. ^b^ Factor loadings between −0.40 and 0.40 are not shown. ^c^ % of variance: percentage of variance explained by each factor.

**Table 6 ijerph-19-13425-t006:** Daily intake (DI*_inh_*) of PAEs (μg/kg-day) and cancer risks resulting from DEHP exposure via inhalation route measured and estimated personal exposure for adults.

Personal Exposure	Measured				Estimated				
	Mean	SD ^a^	Median	95th ^c^	Mean	SD ^a^	Median	95th ^c^	*p*-Value ^b^
DMP (μg/kg-day)	2.2 × 10^−5^	1.8 × 10^−5^	1.6 × 10^−5^	5.5 × 10^−5^	3.8 × 10^−5^	3.9 × 10^−5^	2.9 × 10^−5^	9.5 × 10^−5^	0.006
DEP (μg/kg-day)	4.6 × 10^−4^	5.1 × 10^−4^	3.1 × 10^−4^	1.5 × 10^−3^	7.6 × 10^−4^	7.0 × 10^−4^	5.0 × 10^−4^	2.3 × 10^−3^	0.01
DnBP (μg/kg-day)	4.3 × 10^−3^	4.2 × 10^−3^	3.1 × 10^−3^	1.2 × 10^−2^	7.2 × 10^−3^	5.7 × 10^−3^	5.6 × 10^−3^	1.7 × 10^−2^	0.003
BBP (μg/kg-day)	1.8 × 10^−2^	3.0 × 10^−2^	1.5 × 10^−3^	2.3 × 10^−2^	2.3 × 10^−2^	2.0 × 10^−2^	5.8 × 10^−3^	5.8 × 10^−2^	0.48
DEHP (μg/kg-day)	1.2 × 10^−1^	1.6 × 10^−1^	7.3 × 10^−2^	5.0 × 10^−1^	1.6 × 10^−1^	1.3 × 10^−1^	1.1 × 10^−1^	4.3 × 10^−1^	0.21
DnOP (μg/kg-day)	6.0 × 10^−3^	1.3 × 10^−2^	1.1 × 10^−3^	3.9 × 10^−2^	4.7 × 10^−3^	7.8 × 10^−3^	2.1 × 10^−3^	2.0 × 10^−2^	0.55
∑_6_PAEs ^d^ (μg/kg-day)	1.4 × 10^−1^	2.0 × 10^−1^	7.6 × 10^−2^	6.2 × 10^−1^	1.7 × 10^−1^	1.6 × 10^−1^	1.2 × 10^−1^	4.9 × 10^−1^	0.35
CR*_inh_* for DEHP ^e^	1.3 × 10^−6^	1.8 × 10^−6^	6.6 × 10^−7^	4.9 × 10^−6^	1.2 × 10^−6^	1.1 × 10^−6^	7.1 × 10^−7^	3.3 × 10^−6^	0.76
CR for DEHP ^f^	2.1 × 10^−5^	3.0 × 10^−5^	1.6 × 10^−5^	8.0 × 10^−5^	2.0 × 10^−5^	13.8 × 10^−5^	1.2 × 10^−5^	5.5 × 10^−5^	0.76

Notes: ^a^ SD: standard deviation. ^b^ The level of significance was taken as *p*-value < 0.05. ^c^ 95th: the 95th percentile. ^d^ ∑_6_PAEs: the sum of six PAE congeners, including DMP, DEP, DnBP, BBP, DEHP, and DnOP. ^e^ An inhalation cancer potency factor of 0.0084 (mg/kg-day)^−1^ was used (e.g., defined in the California Office of Environmental Health Hazard Assessment Air Toxic Hot Sports Program). ^f^ An oral slope factor of 0.014 (mg/kg-day)^−1^ was used (e.g., U.S. EPA’s Integrated Risk Information System, based on hepatocellular carcinoma and adenoma).

## Data Availability

The data source is from Kin-Fai Ho, Jockey Club School of Public Health and Primary Care, Chinese University of Hong Kong, Hong Kong, China (kfho@cuhk.edu.hk).

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
