# Peer review of "Occurrence and Risk Assessment of Personal PM2.5-Bound Phthalates Exposure for Adults in Hong Kong"

_ijerph, 2022, doi:10.3390/ijerph192013425_

Round 1

Reviewer 1 Report

The manuscript is well written and merits publication. However, the following points need to be addressed before proceeding with the article for publication.

In the introduction section, the authors have provided background to build their arguments however the literature references are not sufficient. The authors should add more literature references. For instance, in lines 57-59, the authors mentioned studies but cited only one study. Please revise. Also, the literature background is not updated. Use recent five years of literature.  

A few abbreviations are not defined in the manuscript. Please revise carefully.

Ethical considerations should have a separate heading and not be merged with any other heading.

Heading 2.4, equation 1, was it adopted from the literature? If so, please provide the reference.

The result part is irrelevantly dragged and should be shrunk to make it precise and comprehensible. This part seems very confusing and is not understandable for its readership.

If PCA is performed, explain it under a separate heading.

There are a few grammatical errors that need to be fixed.

Author Response

Dear Editor and Reviewers: We thank you for taking your valuable time reviewing this manuscript, and we appreciate your comments and suggestions. We have made appropriate modifications to the main text. Our detailed point-to-point responses to the reviewers' comments are given below.

Reviewer #1

The manuscript is well written and merits publication. However, the following points need to be addressed before proceeding with the article for publication.

Response: We thank the reviewer for the positive feedback on this manuscript. We have extensively revised the initial manuscript to address the reviewer's comments and suggestions.

Comment 1: In the introduction section, the authors have provided background to build their arguments however the literature references are not sufficient. The authors should add more literature references. For instance, in lines 57-59, the authors mentioned studies but cited only one study. Please revise. Also, the literature background is not updated. Use recent five years of literature.  

Response: We thank the reviewer for the suggestion. We updated the literature references by citing the recent five years of research findings and made appropriate changes to the Introduction as suggested. (Lines 57, 62, 63, 64, 66, and 82)

Comment 2: A few abbreviations are not defined in the manuscript. Please revise carefully.

Response: We have made appropriate revisions to the main text, e.g., Lines 15-16,19, 182.

Comment 3: Ethical considerations should have a separate heading and not be merged with any other heading.

Response: We thank the reviewer for this comment. The ethical approval was reported in a separate section (Lines 870-872, Lines 873-874).

Comment 4: Heading 2.4, equation 1, was it adopted from the literature? If so, please provide the reference.

Response: We have added the related reference per the reviewer’s suggestion (Line 224).

Comment 5: The result part is irrelevantly dragged and should be shrunk to make it precise and comprehensible. This part seems very confusing and is not understandable for its readership.

Response: We thank the reviewer for this comment. We extensively revised the initial manuscript and made appropriate changes to the main text per the reviewer’s suggestion.

Comment 6: If PCA is performed, explain it under a separate heading.

Response: We have added a separate sub-section to the Methods per the reviewer’s suggestion. (Section 2.6 Source Identification)

Comment 7: There are a few grammatical errors that need to be fixed.

Response: We have thoroughly reviewed the main text and made the appropriate change accordingly.

Reviewer 2 Report

The manuscript entitled "Occurrence and risk assessment of personal PM2.5-bound phthalates exposure for adults in Hong Kong" presents a detailed overview of the contamination of the outdoor and indoor environment by PAEs. The methodology is appropriately chosen and used. Information on potential sources of PAEs and health risks resulting from PM inhalation is a significant benefit.

I make a few comments.

In the introduction, you thoroughly discuss the types, origin and possible adverse effects of PAEs. PAEs are part of particulate matter, also PM2.5, which have their own characteristics, sources and undergo various processes in the air. It would be appropriate to provide some basic information about PM.

Line 111: What does the abbreviation PEM mean?

Lines 211-216: How does PCA analysis work and what does it reveal? I recommend giving at least a basic indication of the purpose of the analysis.

Figure 2: the data on the x-axis (subject ID) is unclear, it is necessary to adjust it accordingly.

Lines 447-506: On what basis were the respective numbers of principal components PC chosen? In PCA, there are as many principal components as there are variables. In this case it would be 6. I recommend putting it in the text.

The text lacks at least basic information about the measured PM2.5 concentrations. What part of PM2.5 do ∑6PAES represent?

Further directions of research could be mentioned in the conclusions.

Author Response

The manuscript entitled "Occurrence and risk assessment of personal PM2.5-bound phthalates exposure for adults in Hong Kong" presents a detailed overview of the contamination of the outdoor and indoor environment by PAEs. The methodology is appropriately chosen and used. Information on potential sources of PAEs and health risks resulting from PM inhalation is a significant benefit. I make a few comments. In the introduction, you thoroughly discuss the types, origin and possible adverse effects of PAEs. PAEs are part of particulate matter, also PM2.5, which have their own characteristics, sources and undergo various processes in the air. It would be appropriate to provide some basic information about PM.

Response: We thank the reviewer for this comment. We have added the following sentence to the Introduction. (Lines 68-70)

Comment 1: Line 111: What does the abbreviation PEM mean?

Response: PEM refers to personal environmental monitor, and we made appropriate changes to the main text. (Line 182)

Comment 2: Lines 211-216: How does PCA analysis work and what does it reveal? I recommend giving at least a basic indication of the purpose of the analysis.

Response: We thank the reviewer for this comment. One of the study scopes is to “investigate the potential sources of PAEs in different exposure metrics” (Line 95)

    PCA has been used for PAEs source identification and is usually applied when there are only six species in previous publications. We added one separate sub-section to the Methods (Section 2.6 Source Identification) per the reviewer’s suggestion.

Comment 3: Figure 2: the data on the x-axis (subject ID) is unclear, it is necessary to adjust it accordingly.

Response: We have made appropriate changes to Figure 2 per the reviewer's suggestion.

Comment 4: Lines 447-506: On what basis were the respective numbers of principal components PC chosen? In PCA, there are as many principal components as there are variables. In this case it would be 6. I recommend putting it in the text.

Response: We thank the reviewer for this comment. PCA has been used in previous publications for PAEs source identification under the condition that only six major congeners were included.

    We have added the following sentences to the main text: “Next, we applied principal component analysis (PCA) to identify the potential sources of PAEs in personal exposure, residential indoor, and outdoor PM2.5. This method has been used for indoor and outdoor PAEs source identification (Chen et al. 2018), and personal exposure source apportionment in previous investigations (Chen et al. 2017). (Section 2.6)

    The following sentences have been added to the main text: “In this study, we employed Varimax normalized rotation to minimize or maximize the loading factors of included species for each rotated principal component. The principal components with eigenvalues greater than one were retained, and factor loadings less than 0.4 were omitted to facilitate source identification. PCA was performed by using IBM SPSS Statistics (Version 26.0).” (Lines 298-301)

Comment 5: The text lacks at least basic information about the measured PM2.5 concentrations. What part of PM2.5 do ∑6PAES represent?

Response: We thank the reviewer for the comment. We have added the measured PM2.5 exposure to the main text. (Line 265)

    In addition, the following sentence was added to the Introduction. “Characteristics of common air pollutants and air toxics [e.g., benzene, formaldehyde, volatile organic compounds, vinyl chloride, metals, and persistent organics] have been investigated in previous studies (Strum and Scheffe 2016; Yang et al. 2018).” (Lines 68-70)

Comment 6: Further directions of research could be mentioned in the conclusions.

Response: We thank the reviewer for the suggestion. The following sentences were added to the Discussion.

    “Additional exploration concerning characteristics of the particle- and gas-phase PAE congeners in different exposure metrics may provide information about exposure errors introduced by sampling methods.” (Lines 835-838)

    “Our results raised concerns over the elevated inhalation risk of exposure to particle-bound DEHP for Hong Kong residents. These results can help facilitate future regulatory action and evidence-based strategies to reduce PAE exposures and protect vulnerable sub-populations (e.g., infants and young children).” (Lines 857-861)

Reviewer 3 Report

The paper attempts to connect risk assessment for disease with PM2.5 bound phthalate particulates. The connections, however, do not include risk due to the presence of other concurrent air contaminants. Though it is well established that phthalates in air are toxic, it also well known that numerous other air pollutants are toxic. Accordingly other air pollutants should also be measured when phthalates are. Questions that should be addressed include:

a) Were increases in phthalates associated with increases in other air pollutants?

b) Are the health effects of the other pollutants the same or different from those of phthalates?

The paper also fails to address the toxicity of mixtures of phthalates and other pollutants present in the sampled air. Mixtures are well known to induce enhanced toxic effects and effects at lower concentration levels than those of the individual components of such mixtures. 

It is very possible that the connection between disease risk and airborne phthalates obtains, but the effects of other pollutants present must be factored into the equation.

Author Response

Reviewer #3

The paper attempts to connect risk assessment for disease with PM2.5 bound phthalate particulates. The connections, however, do not include risk due to the presence of other concurrent air contaminants. Though it is well established that phthalates in air are toxic, it also well known that numerous other air pollutants are toxic. Accordingly other air pollutants should also be measured when phthalates are. Questions that should be addressed include:

a) Were increases in phthalates associated with increases in other air pollutants?

Response: We thank the reviewer for this comment. Study results concerning other chemical components of PM2.5 [elemental carbon (EC), organic carbon (OC), polycyclic aromatic hydrocarbons (PAHs)] were reported in previous findings (Chen et al. 2020). Results concerning the correlation between phthalates with personal PM2.5 exposure were also reported, and weak correlations were shown between particle-bound phthalates and PM2.5 concentrations in personal exposure (r = 0.07), indoor (r = 0.21) and outdoor (r = -0.14).

b) Are the health effects of the other pollutants the same or different from those of phthalates?

Response: We thank the reviewer for this comment. Different health effects were attributable to personal exposure to PM2.5 components. Our previous publication in Fan et al. (2018) investigated the associations between personal PM2.5 exposure and respiratory inflammation and assessed the effects of various PM2.5 components (OC, EC, PAHs, phthalates). Moderated correlations were shown between phthalates and oxy-PAHs (r = 0.49). Weak correlations were found between phthalates with PM2.5 concentration (r = 0.07), OC (r = 0.18), EC (0.09), parent-PAHs (r = 0.19), respectively. The results derived from the mixed effects model showed that among the constituents investigated, carcinogenic PAHs were attributed to airway inflammation in study participants.

c) The paper also fails to address the toxicity of mixtures of phthalates and other pollutants present in the sampled air. Mixtures are well known to induce enhanced toxic effects and effects at lower concentration levels than those of the individual components of such mixtures. 

Response: We thank the reviewer for this comment. We listed the research objectives of this study in the Introduction: “The research objectives of this study are to (1) examine the occurrence, concentrations, and variations of PM2.5-bound PAE congeners (i.e., DMP, DEP, DnBP, BBP, DEHP, and DOP) in personal exposure and residential indoors; (2) characterize the within- and between-individual (or home) variability of PAE congeners in personal exposure and residential indoor; (3) investigate the potential sources of PAEs in different exposure metrics, and (4) assess health risks caused by inhalation exposure to DEHP for Hong Kong adults based on directly measured exposure and estimated personal exposure.” (Lines 85-91)

    The results concerning the toxicity effects of PM2.5 components were reported in a separate publication (Chen et al. 2020).

d) It is very possible that the connection between disease risk and airborne phthalates obtains, but the effects of other pollutants present must be factored into the equation.

Response: We thank the reviewer for this comment. As stated in previous responses, phthalates demonstrated distinct characteristics compared to other PM2.5 components, for instance, showing a weak correlation with PM2.5 concentrations, revealing different patterns in various exposure types, particle-bound phthalates were the highest in ambient air compared to other exposure categories. This investigation focused on the particle-bound phthalates, the health risks of PAHs (Chen et al. 2022), and the toxicity effects of individual hazardous components and the mixture can be referred to in the previous publication (Chen et al. 2020).

Round 2

Reviewer 3 Report

The improvements made still do not address the issues raised in the initial review.

Author Response

We thank the reviewer for taking the time to review this manuscript, and we appreciate your comments and suggestions. We have made appropriate changes to the main text.

  1. Were increases in phthalates associated with increases in other air pollutants?

Response: The following sentences were added to the main text. “The results showed weak correlations between ∑6PAEs and PM2.5 concentrations in different exposure categories [e.g., personal exposure (r = 0.07), indoor (r = 0.21) and outdoor (r = -0.14)]; weak correlations were found for ∑6PAEs with organic carbon (r = 0.18), elemental carbon (r = 0.09), and ∑parent-PAHs (r = 0.19), respectively (Chen et al. 2020; Fan et al. 2018).” (Lines 626-630)

  1. Are the health effects of the other pollutants the same or different from those of phthalates?

Response: We thank the reviewer for this comment. Associations of organic carbon (OC), elemental carbon (EC), PAHs, and phthalate in personal PM2.5 exposures with airway inflammation were investigated in our previous publication. The results showed that among the PM2.5 components (OC, EC, PAHs, phthalates) examined, combusted-derived pollutants are dominant factors contributing to airway inflammation in healthy adults of Hong Kong (Fan et al. 2018).

      The following sentences were added to the main text. “Different health effects were attributable to specific components of PM2.5 (Yang et al. 2018). To put measured and estimated values in the context of related health risks, we assessed inhalation risks of exposure to particle-bound DEHP due to its carcinogenetic potential.” (Lines 808-811)

  1. The paper also fails to address the toxicity of mixtures of phthalates and other pollutants present in the sampled air. Mixtures are well known to induce enhanced toxic effects and effects at lower concentration levels than those of the individual components of such mixtures. 

Response: We thank the reviewer for this comment. This campaign investigated two specific air toxics (phthalates and PAHs). The cancer risks of PAHs were reported in a previous publication (Chen et al. 2022). (Lines 209-210)

    The International Agency for Research on Cancer and the United States Environmental Protection Agency (U.S. EPA) classified DEHP and BBP as possible human carcinogens (Group B2 and Group C) based on experimental animal studies (Caldwell 2012; U.S. EPA 1987). As we have stated in lines 264-265, other PAE congeners were not included in the risk assessment because their cancer slope factors are not available. (Lines 260-265)

    The results concerning the in vitro toxicity effects of PM2.5 components (organic carbon, elemental carbon, PAHs, phthalates) were reported in a separate publication (Chen et al. 2020).”  

  1. It is very possible that the connection between disease risk and airborne phthalates obtains, but the effects of other pollutants present must be factored into the equation.

Response: We thank the reviewer for this comment. “… PAEs are the most abundant endocrine-disrupting chemical compounds in ambient air” (Lines 70-71). Our previous investigation in personal exposure showed significant associations for PAHs with airway inflammation but not for PAEs, suggesting the target organ may differ across PM2.5 constituents.

      We understand that it is advantageous to include different air toxics in our research. However, only particle-bound phthalates and PAHs were measured from the filter samples. Weak correlations were shown between ∑PAEs and ∑parent-PAHs (r = 0.19). This investigation focused on particle-bound phthalates (Lines 91-96).